# The Filippi’s Glands of Giant Silk Moths: To Be or Not to Be?

**DOI:** 10.3390/insects12111040

**Published:** 2021-11-19

**Authors:** Hana Sehadova, Radka Zavodska, Michal Zurovec, Ivo Sauman

**Affiliations:** 1Biology Centre of the Czech Academy of Sciences, Institute of Entomology, 370 05 Ceske Budejovice, Czech Republic; sehadova@entu.cas.cz (H.S.); radkaz@pf.jcu.cz (R.Z.); zurovec@entu.cas.cz (M.Z.); 2Faculty of Science, University of South Bohemia, 370 05 Ceske Budejovice, Czech Republic; 3Faculty of Education, University of South Bohemia, 370 05 Ceske Budejovice, Czech Republic

**Keywords:** Filippi’s glands, Saturniidae, Sphingidae, Pyralidae, *Bombyx mori*, silk

## Abstract

**Simple Summary:**

It is well documented that silkworms (Saturniidae) possess silk glands’ accessory glandular organs, the Filippi’s glands (FGs). These accessory glands are believed to play an important role in the cocoon construction. Unexpectedly, we have identified several silk moth species that are completely missing the FGs during the entire larval development and still produce fully functional cocoons. This finding suggests that the role of FGs is not crucial for the cocoon spinning.

**Abstract:**

The Filippi’s glands (FGs), formerly “Lyonet’s glands”, are paired accessory organs associated with the silk glands. They are unique to Lepidoptera caterpillars and their exact role is not clear. The FGs are thought to be involved in the construction of a silk cocoon in bombycoid moths. FGs can differ in size and shape, therefore, in this study we attempt to find a correlation between FG morphology and phylogenetic position within the Bombycoidea. We use light and electron microscopy to examine the presence and morphology of FGs in a range of wild (giant) silk moths and several related species. Our results confirm that the majority of studied silk moth species have complex type of FGs that continuously increase in size during larval development. We identified several species of giant silk moths and two hawk moth species that completely lack FGs throughout their larval development. Finally, in several hawk moth species in which FGs are well developed during the first larval stage, these glands do not grow and remain small during later larval growth. Our results suggest that FGs are not critical for spinning and that loss of FGs occurred several times during the evolution of saturniids and sphingids. Comparison of FGs in different moths is an important first step in the elucidation of their physiological significance.

## 1. Introduction

The main structures responsible for cocoon construction are the silk glands (SGs), which represent modified labial glands and form a large portion of the caterpillar body mass. The SGs are associated with paired small accessory glands, called the Filippi’s glands (FGs). The FGs were first described in a noctuid moth, *Cossus cossus*, and originally referred to as “Lyonet’s glands,” after their discoverer. In the silk moth, *Bombyx mori*, however, they were found by Filippi and named accordingly [1].

It has been suggested that FGs may play several important roles in the process of maturation of silk material in the anterior silk glands (ASGs) as well as in the process of silk spinning. FGs have been associated with the production of lubricant for silk fibers [2,3], transport of small solutes (ions, sugars, and amino acids) to ASGs [4], and the biosynthesis of fatty acids [5] and lipids [6]. However, the exact role of FGs in silk filament formation and silk spinning has not been fully elucidated. Since the complete removal of FGs in B. mori had no obvious effect on silk spinning behavior or silk quality [7,8], the role of FGs in cocoon spinning is probably only supplementary. 

The FGs may differ in size and shape. Based on their gross morphology, FGs were classified into two major types. The first type appears as simple glandular lobes arising dorsally from the anterior part of the SGs (ASGs), without any apparent gland canals. The second type has well-developed leaf-like or tubular lobes and is located on the ventrolateral sides of the esophagus, close to the subesophageal ganglion. These FGs are connected with the lumen of the SGs by proximal tubular ducts entering dorsally into the ASGs [1].

The first type of FG was described in diverse lepidopteran families in the following species: Cossidae, *C. cossus*; Tortricidae, *Grapholita funebra*; Erebidae, *Adris tyrannus amurensis*, *Calliteara pudibunda*, and *Hyphantria cunea*; Sphingidae, *Herse convolvili, Sphinx ligustri,* and *Mimas tiliae*; Papilionidae, *Papilio Xuthus* [9]; Nymphalidae, *Aglais urticae*; and Pieridae, *Pieris rapae*, *Pieris napi*, and *Pieris brassicae* [1,9]. The second, more complex type was found in the families Bombycidae, *B. mori* [1,9,10]; Saturniidae, *Antheraea mylitta* and *Antheraea pernyi* [1,9]; Lasiocampidae, *Macrothylacia rubi* [1]; Noctuidae, *Leucania separata* [9], *Helicoverpa zea* [11], *Helicoverpa armigera* [12], *Spodoptera littoralis* [13], *Spodoptera frugiperda* [14], *Melanchra persicariae*, and *Colocasia coryli* [1]; Notodontidae, *Cerura vinula* and *Phalera bucephala* [1]; Crambidae, *Diatraea saccharalis* [15] and *Ostrinia nubilalis* [16]; Erebidae, *Orgyia antiqua*, *Spilosoma lubricipeda*, *Spilosoma urticae*, *Atolmis rubricollis*, and *Lymantria monacha* [1]; and Nymphalidae, *Inachis io* [1].

The economically most important silk producers belong to two families of Bombycoidea—Bombycidae and Saturniidae. The caterpillars of these moths spin spectacular silk-like cocoons that serve as a protective covering for the pupal stage. Members of these families have also been found to have well-developed FGs. Most members of a sister group of silk moths, the hawkmoths (Sphingidae), such as the tobacco hawkmoth, *Manduca sexta* [17], and the privet hawkmoth, *S. ligustri* [1], also possess FGs, although they do not spin cocoons. However, during the first larval instar, they do produce silk fibers to attach themselves to the food plant [18,19,20,21,22].

In the course of our previous study [23], we had noticed the absence of FGs in the larvae of the fifth instar of the cecropia silk moth, *H. cecropia*. To determine whether the absence of FGs is rather an exception in the superfamily Bombycoidae, we studied the morphology, anatomy, and development of FGs in selected silk moth and hawk moth species. We selected representatives of major saturniid and sphinghid clades based on the current phylogenic tree [24]. We show here that FGs are absent in a number of saturniid species, despite their ability to produce complex functional cocoons. In contrast, FGs are commonly present in hawkmoths, despite their inability to spin cocoons. A comparison of the selected species of saturniids and sphingids allows us to search for correlations between FG morphology and their phylogenetic position and to speculate on their function.

## 2. Materials and Methods

### 2.1. Animals

Larvae from the greater wax moth, *Galleria mellonella*, Mediterranean flour moth, *Ephestia kuehniella*, and commercial silkworm, *B. mori*, were obtained from our laboratory colonies and reared on their respective standard artificial diets as described elsewhere [25,26,27]. The eggs of the silk moth and hawk moth species used in this study were purchased from Worldwide Butterflies Ltd. (Dorset, UK, http//:www.wwb.co.uk, accessed on 15 November 2021).

After hatching, the larvae were reared in a standard insect-rearing facility, fed on freshly cut branches of their respective food plants [28,29], which were as follows: for the African death’s-head hawk moth, *Acherontia atropos*—Korean privet (*Ligustrum ovalifolium*); Indian moon moth, *Actias selene*—weeping willow (*Salix babylonica*); tau emperor moth, *Aglia tau*—small-leaved lime (*Tilia cordata*); Chinese oak silk moth, *Antheraea pernyi*—common oak (*Quercus robur*)*;* Polyphemus moth, *Antheraea polyphemus*—common oak (*Quercus robur*); Japanese oak silk moth, *Antheraea yamamai*—common oak (*Quercus robur*); Suraka silk moth, *Antherina suraka*—Korean privet (*Ligustrum ovalifolium*); Atlas moth, *Attacus atlas*—tree of heaven (*Ailanthus altissima*) or cherry laurel (*Prunus laurocerasus*); promethea silk moth, *Callosamia promethea*—tulip tree (*Liriodendron tulipifera*); oleander hawk moth, *Daphnis nerii*—Korean privet (*Ligustrum ovalifolium*); cecropia silk moth, *Hyalophora cecropia*—wild cherry (*Prunus avium*); bedstraw hawk moth, *Hyles gallii*—hedge bedstraw (*Galium mollugo*); poplar hawk moth, *Laothoe populi*—goat willow (*Salix caprea*); lime hawk moth, *Mimas tiliae*—lime tree (*Tilia cordata*); African emperor moth, *Nudaurelia krucki*—cherry laurel (*Prunus laurocerasus*); ailanthus silk moth, *Samia cynthia*—tree of heaven (*Ailanthus altissima*); small emperor moth, *Saturnia pavonia*—goat willow (*Salix caprea*); Ligurian emperor moth, *Saturnia pavoniella*—goat willow (*Salix caprea*); giant peacock moth, *Saturnia pyri*—wild cherry (*Prunus avium*); eyed hawk moth, *Smerinthus ocellatus*—goat willow (*Salix caprea*); and privet hawk moth, *Sphinx ligustri*—Korean privet (*Ligustrum ovalifolium*).

### 2.2. Light Microscopy

The ASGs with/without the FGs attached were dissected from the water-anesthetized larvae under Ringer’s solution (pH 7.4), fixed in 4% paraformaldehyde for at least 30 min, and then washed in phosphate-buffered saline (PBS). The glands were transferred to a drop of PBS on a microscopy slide, covered with a coverslip, and imaged under an Olympus BX51 microscope equipped with a charge-coupled device (CCD) camera (Olympus DP80). The final photographs were reconstructed by stitching several Z-stack images. Larger tissues were dehydrated in an ethanol series (50%, 70%, 90%, and 100%) for 15 min each. To the 100% ethanol, the identical volume of methyl salicylate was added. After the ethanol evaporated, the tissues were stored and mounted in methyl salicylate to clear the tissue for imaging. We used at least five experimental animals for each species examined.

### 2.3. Masson Trichrome Staining

The heads of the water-anesthetized larvae were fixed overnight at 4 °C in Bouin-Hollande solution without acetic acid but supplemented with mercuric chloride [30]. Standard techniques were used for tissue dehydration, embedding in paraplast (Sigma-Aldrich, Inc., St. Louis, MO, USA), sectioning at 7–10 μm, deparaffinization, and rehydration. The sections were treated with Lugol’s iodine followed by a 7.5% solution of sodium thiosulphate to remove residual heavy metal ions and then washed in distilled water. Staining was performed with the HT15 Trichrome Stain (Masson Stain) Kit (Sigma-Aldrich, Inc., St. Louis, MO, USA) according to the manufacturer’s protocol. Stained sections were dehydrated, mounted in dibutylphthalate polystyrene xylene (DPX) mounting medium (Sigma-Aldrich, Inc., St. Louis, MO, USA), and viewed and imaged under an Olympus BX51 microscope with CCD camera (Olympus DP80, Olympus Corporation, Tokyo, Japan).

## 3. Results

### 3.1. Absence of Filippi’s Glands in Some Giant Silk Moth Species (Saturniidae)

We examined the larvae of fourteen selected representatives of main clades of the family Saturniidae (Figure 1). The entire spinning apparatuses of the last instar larvae were dissected and investigated under the light microscope. The presence or absence of FGs was verified on the cross-sections of the spinning apparatus labeled by Masson trichrome stain.

In representatives of the genera Antheraea (*A. pernyi, A. polyphemus,* and *A. yamamai*), Actias (*A. selene*), Saturnia (*S. pavonia, S. pavoniella,* and *S. pyri*), and Antherina (*A. suraka*), well-developed FGs with globular lobes located on the ventrolateral side of each paired SG were found. The canaliculi of the glandular lobes converged into the duct running anteriorly and entered the SG dorsally near their junction at the posterior part of the silk press (Figure 2A–C, Appendix A). Interestingly, even within one silk moth genus such as Antheraea, we observed considerable variability in the morphology and anatomy of the FGs. While *A. polyphemus* and *A. pernyi* have well-developed globular FGs with long efferent glandular ducts (Figure 2A, Appendix A), the FGs of the sibling species, *A. yamamai*, had a branched and rather tubular structure, where the canaliculi of the individual tubules converged to a common duct (Appendix A).

Compared to *A. polyphemus*, species from the genera Actias and Saturnia had FGs of a similar size, but the glandular ducts were noticeably shorter (Figure 2B,C and Appendix A). In *A. suraka*, the FGs were significantly smaller with very short ducts, so their lobes were close to the SGs (Appendix A).

Several phylogenetically closely related silk moth species from the same subfamily (Saturniinae)—namely, the genera Hyalophora (*H. cecropia*), Callosamia (*C. promethea*), Samia (*S. cynthia*), and Attacus (*A. atlas*)—did not possess FGs at all. Two other more distant genera of saturniid moths, the Nudaurelia (*N. krucki*) and Aglia (*A. tau*), were also missing FGs (Figure 2D–F, Appendix A).

### 3.2. Development of Filippi’s Glands in Saturniidae

To investigate the presence and development of FGs during larval growth, we examined the larvae of the first and third instars of most of the above-mentioned species (Figure 3, Figure 4, Appendix A). In all species that possessed the FGs in the fifth larval instar, the FGs were well established already in the first instar immediately after egg hatching (Figure 3A,C,E, Appendix A) and continued to develop thereafter, as indicated by their increasing size in the third (Figure 3B,D,F, Appendix A) and last larval instars (Figure 2A–C, Appendix A). The species without FGs in their last larval instars lacked them throughout the entire larval development (Figure 4).

### 3.3. Heterogeneity of Filippi’s Glands Occurrence in Hawk Moths (Sphingidae)

Based on published data from the tobacco hornworm, *M. sexta*, a representative of the silk moth sister group, the hawk moths (Sphingidae), in which the FGs were developed and functional during the first instar and disappeared in later larval instars [17], we investigated the presence of FGs in seven additional species from the Sphingidae family (Figure 5 and Figure 6). Similarly to *M. sexta*, it was observed that the oleander hawk moth, *D. nerii*, bedstraw hawk moth, *H. gallii*, African death’s-head hawk moth, *A. atropos*, privet hawk moth, *S. ligustri*, and eyed hawk moth, *S. ocellatus*, all had well defined FGs in their first instar larvae (Figure 5A and Figure 6A,D,G,J,M). However, in contrast to *M. sexta* [17], the rudiments of the FGs in all these species were found to persist throughout the entire larval development to the end of the last larval instar (Figure 5A–C and Figure 6). The FGs did not grow larger from the first to last larval instars, so the size of the FGs becomes disproportional when compared to the size of the growing and developing SGs. All the above-mentioned species produce functional silk fibers only during the first larval instar (Appendix A).

Another two hawk moth species investigated, the lime hawk moth, *M. tiliae*, and poplar hawk moth, *L. populi*, did not possess even rudimentary FGs during the entire larval development (Figure 5D–F) and do not produce any silk fibers even in the first larval instar.

### 3.4. Filippi’s Glands in Other Bombycoidea Species

The commercial silk moth *B. mori* is a representative of the third family (*Bombycidae*) of the superfamily Bombycoidea. Due to the lack of anatomical data of FGs in *B. mori*, we investigated the anatomy and morphology of its FGs by light microscopy at the levels of both whole mounts and histological sections of the spinning apparatus (Figure 7A,B). The arrangement of the FGs was similar to that observed in the Saturniidae species. The *B. mori* FGs have the same localization on the ventrolateral side of the SGs, but the morphology of the secretory lobes has a rather leafy structure compared to the globular architecture of the FGs in Saturniidae.

### 3.5. Filippi’s Glands in Some Ancestral Lepidopteran Species

To compare the morphology and anatomy of FGs in *Bombycoidea* with more ancestral lepidoperan species we investigated FGs in two representatives from the family Pyralidae: the greater wax moth, *Galleria mellonella* (Figure 7C,D), and the Mediterranean flour moth, *Ephestia kuehniella* (Figure 7E,F). In these two ancestral lepidopteran species, the FGs were found to be represented by simple invagination of the SG epithelium without any obvious duct connections with the ASGs.

## 4. Discussion

In this study, we have examined the anatomy and morphology of the spinning apparatus of fourteen giant silk moth species (Figure 1). In eight species (*A. selene*, *A. pernyi*, *A. polyphemus*, *A. yamamai*, *A. suraka*, *S. pavonia*, *S. pavoniella*, and *S. pyri*), we have identified well-developed FGs with a similar localization and comparable basic morphology (Figure 2A–C, Appendix A). The FGs were located on the ventrolateral side of the SGs, in close proximity to the suboesophageal ganglion. Concerning the detailed morphology, all the above-mentioned species have globular-type FGs with efferent ducts that open into each SG, but they differ in terms of the size and shape of the glandular lobes and the length of the connecting ducts. In all these species, we were able to detect the FGs immediately after egg-hatching, and to observe their continuous development through all larval stages. Thus, it is clear that the FGs are established during the late embryonic development at the stage of the pharate larva.

We have also identified six silk moth species in which the FGs are completely missing: *A. atlas, A. tau*, *C. promethea*, *H. cecropia*, *N. krucki*, and *S. cynthia* (Figure 2D–E and Appendix A). In all these species, the FGs are absent throughout the entire larval development. The absence of FGs was previously reported in *Philosamia cynthia ricini* [9,31]. Most of the saturniid species from the subfamily Saturniinae, which lack FGs, construct fully functional cocoons. One species from the subfamily Bunaeinae, *N. krucki*, was also found lacking the FGs, but it does not construct the cocoon at all and pupates underground similarly to sphingid moths [29,32].

The oldest ancestral saturniid species investigated in the current study, *Aglia tau* (Saturniidae, subfamily Agliinae), also did not possess FGs. Caterpillars of this species form sparse cocoons, pupation occurs on the ground surface, and the pupa actively exits the cocoon prior to adult eclosion [33].

Like the giant silk moths (Saturniidae), several species of their sister group, the hawk moths (Sphingidae), possess FGs and produce functional adhesive fibers during the first larval instar that are necessary for attachment to the food plant as fall protection [17]. The FGs in hawk moths persist throughout larval development until the final instar. However, unlike saturniids, their FGs do not increase in size during larval development as do the labial glands, and remain only in a rudimentary form until pupation (Figure 3 and Figure 6A–L).

We also identified two species of hawk moths without FGs: the lime hawk moth, *M. tiliae*, and the poplar hawk moth, *L. populi* (Figure 5D–F and Figure 6M–O). Although the existence of FGs in *M. tiliae* was reported previously [1], our careful microscopic reexamination revealed no FGs in this species throughout larval development. Coincidentally, the two above-mentioned species do not spin any silk fibers during the first larval instar. It is noteworthy that a closely related hawk moth species, *S. ocellatus*, from the same subfamily (Smerinthinae) has well-developed FGs during the first instar and produces rescue silk fibers. Thus, the presence of FGs in hawk moths seems to correlate with the production of rescue silk filaments in first instar larvae.

Considering our results from the phylogenetic view of the superfamily Bombycoidea, it is evident that the FGs in this taxon were independently lost several times during its evolution. The secondary loss of the FGs is supported by the fact that FGs are present in many basal lepidopteran species, such as *E. kuehniella* and *G. mellonella* (Pyralidae). It is, therefore, very likely that the most ancestral lepidopteran species of the Bombycoidea superfamily (Bombycidae, Sphingidae, and Saturniidae) possessed well developed FGs. The relatively frequent loss of FGs in some saturniid and sphingid species further supports the hypothesis that the FGs are not crucial for cocoon construction and probably play some minor role in maturation of silk fibers and/or resulting cocoon architecture. To decipher the exact function of FGs will require future additional molecular, proteomic, and metabolomic studies.

## 5. Conclusions

Some giant silk moth species do not possess Filippi’s glands during the entire larval development.

The Filippi’s glands are not essential for cocoon construction in the giant silk moths.

In hawk moths, the presence of Filippi’s glands correlates with the production of rescue silk fibers in the first instar larvae.

FGs in this taxon were independently lost several times during its evolution.

## Figures and Tables

**Figure 1 insects-12-01040-f001:**
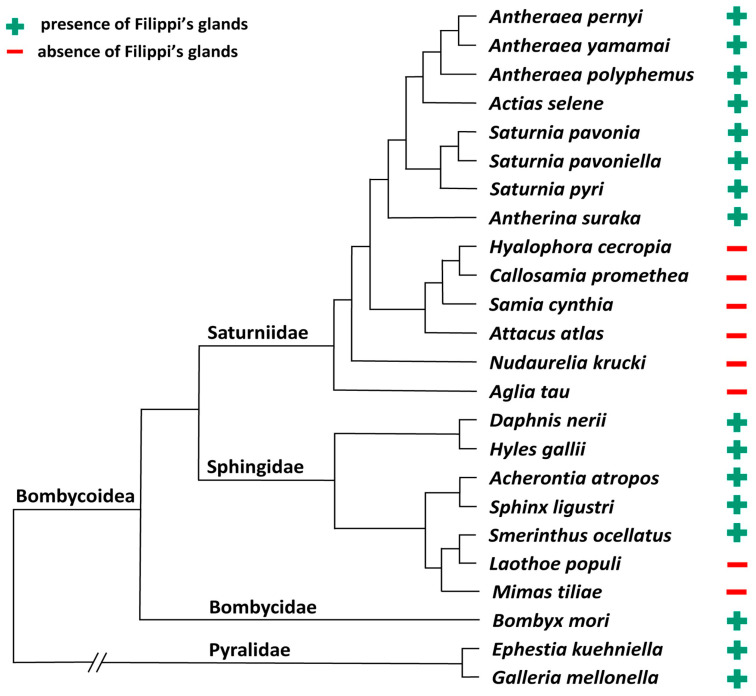
Simplified phylogenetic tree of Bombycoidea superfamily with respect to the species used in this study. The family Pyralidae serves as an outgroup [24].

**Figure 2 insects-12-01040-f002:**
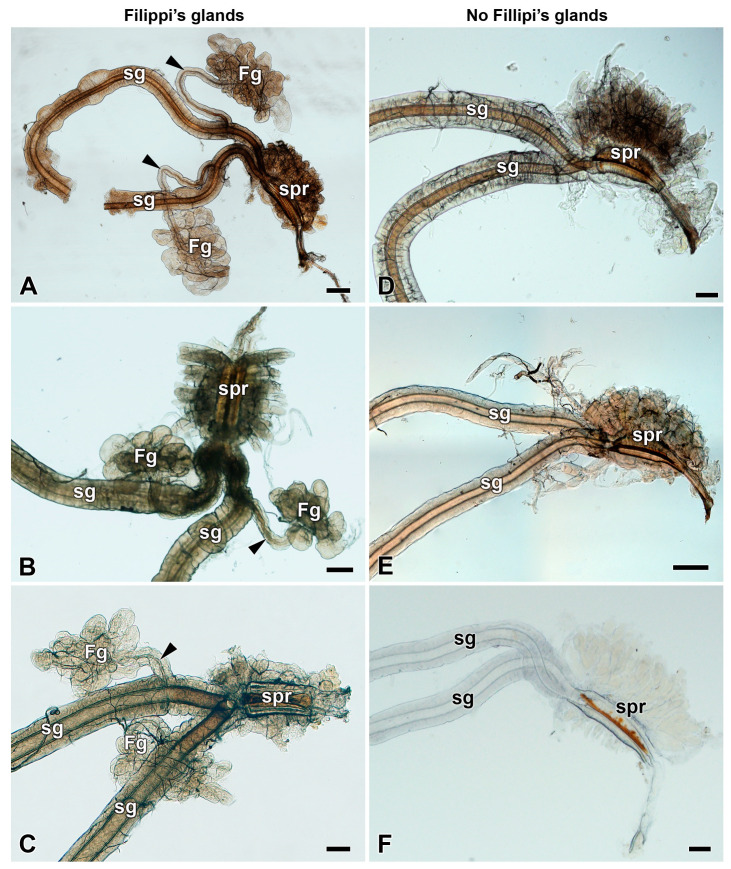
Spinning apparatuses of the 5th larval instar of Saturniidae. (**A**) The polyphemus moth, *Antheraea polyphemus.* (**B**) The Indian moon moth, *Actias selene.* (**C**) The small emperor moth, *Saturnia pavonia.* (**D**) The cecropia silk moth, *Hyalophora cecropia.* (**E**) The Eri silk moth, *Samia cynthia.* (**F**) The Atlas moth, *Attacus atlas.* Note that the FGs in *H. cecropia*, *S. cynthia*, and *A. atlas* are completely missing. The arrowheads indicate the ducts of the FGs entering the lumen of the SGs. Abbreviations: Filippi’s gland (Fg), silk gland (sg), and silk press of the spinning apparatus (spr). Scale bars: 100 µm.

**Figure 3 insects-12-01040-f003:**
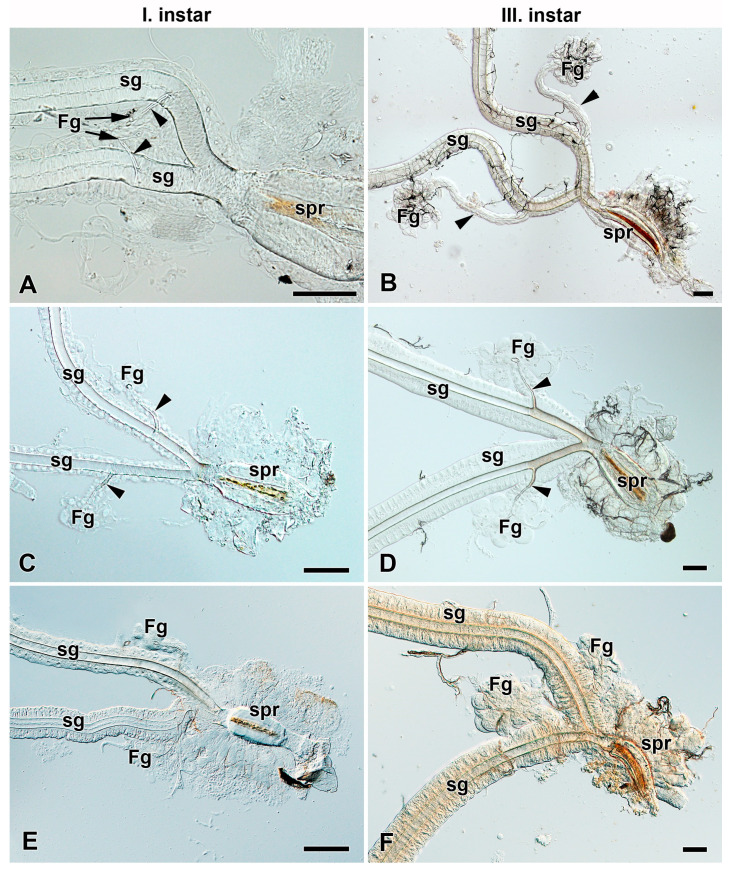
Spinning apparatuses with Filippi’s glands in Saturniidae larvae of the first and third instars. (**A**,**B**) The Polyphemus moth, *Antheraea polyphemus.* In (**A**), the FGs are shown by arrows. (**C**,**D**) The Indian moon moth, *Actias selene.* (**E**,**F**) The small emperor moth, *Saturnia pavonia.* The arrowheads indicate the ducts of the FGs entering the lumen of the SGs. Abbreviations: Filippi’s gland (Fg), silk gland (sg), and silk press of the spinning apparatus (spr). Scale bars: 100 µm.

**Figure 4 insects-12-01040-f004:**
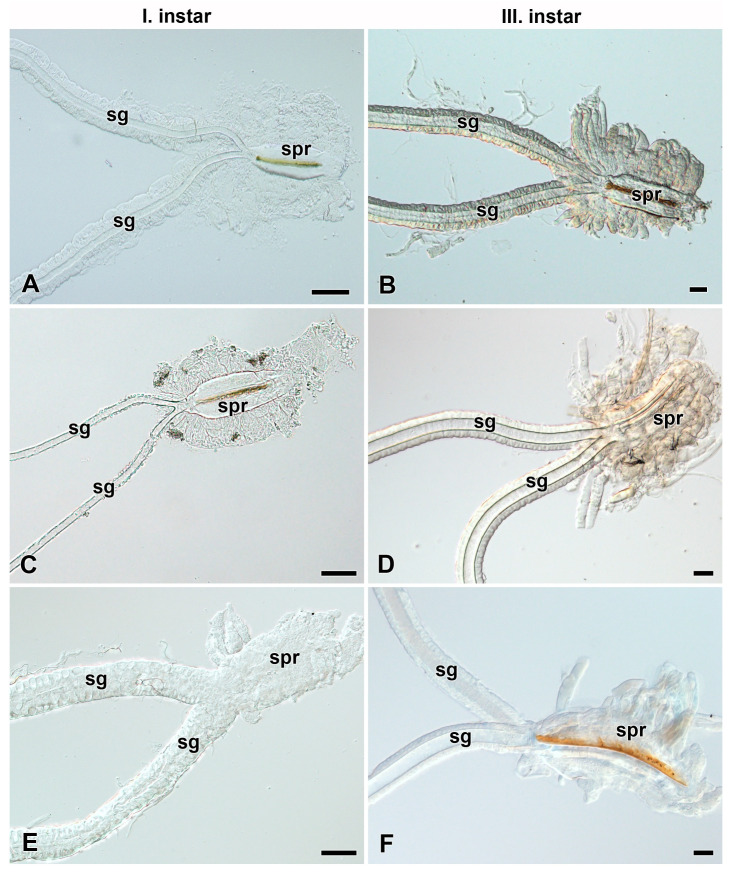
Spinning apparatuses without Filippi’s glands in Saturniidae larvae of the first and third instars. (**A**,**B**) The cecropia silk moth, *Hyalophora cecropia.* (**C**,**D**) The Eri silk moth, *Samia cynthia.* (**E**,**F**) The Atlas moth, *Attacus atlas.* Abbreviations: Filippi’s gland (Fg), silk gland (sg), and silk press of the spinning apparatus (spr). Scale bars: 100 µm.

**Figure 5 insects-12-01040-f005:**
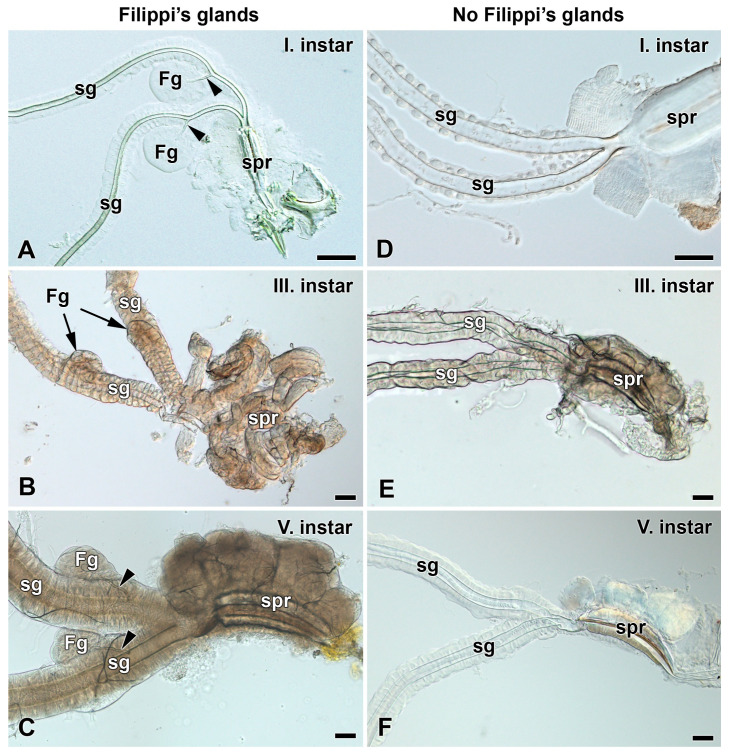
Development of spinning apparatuses of two representative species of hawk moth larvae (Sphingidae), first, third, and fifth instars. (**A**–**C**) Brightfield microscopy images of the spinning apparatus of the oleander hawk moth, *Daphnis nerii*. The arrowheads indicate the ducts of the FGs entering the lumen of the SGs. The arrows depict the FGs. (**D**–**F**) The spinning apparatus of the lime hawk moth, *Mimas tiliae*. Note that the FGs are completely missing through the entire larval development. Abbreviations: Filippi’s gland (Fg), silk (labial) gland (sg), and silk press of the spinning apparatus (spr). Scale bars: 50 µm.

**Figure 6 insects-12-01040-f006:**
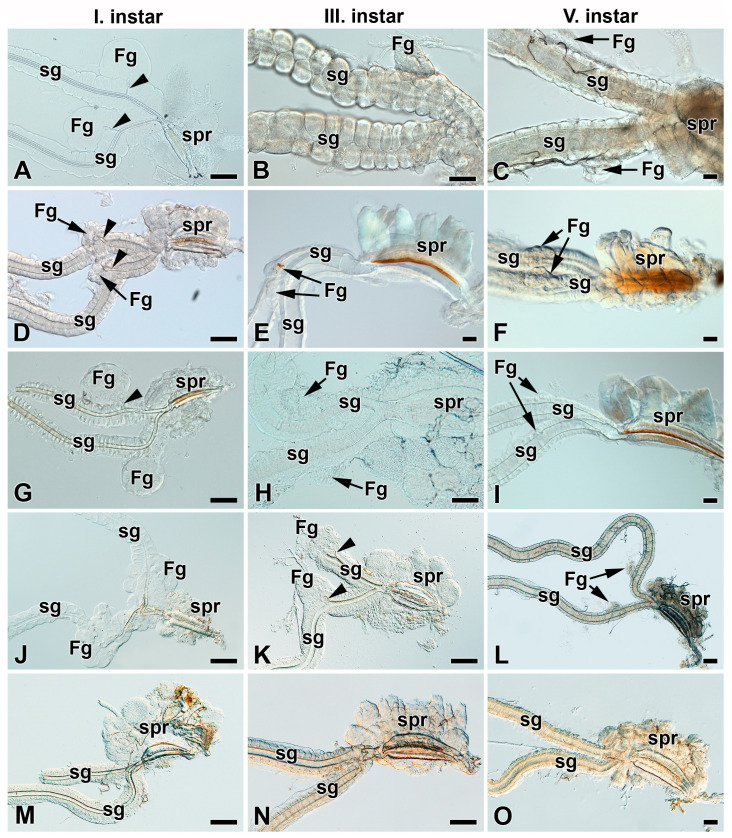
Development of spinning apparatuses of Sphingidae, first, third, and fifth larval instars. (**A**–**C**) Bedstraw hawk moth, *Hyles gallii.* (**D**–**F**) Death’s-head hawk moth, *Acherontia atropos*. (**G**–**I**) Privet hawk moth, *Sphinx ligustri.* (**J**–**L**) The eyed hawk moth, *Smerinthus ocellatus*. (**M**–**O**) The poplar hawk moth, *Laothoe populli*. The FGs are shown by arrows. The arrowheads indicate the ducts of the FGs entering the lumen of the SGs. Abbreviations: Filippi’s gland (Fg), silk gland (sg), and silk press of the spinning apparatus (spr). Scale bars: 50 µm.

**Figure 7 insects-12-01040-f007:**
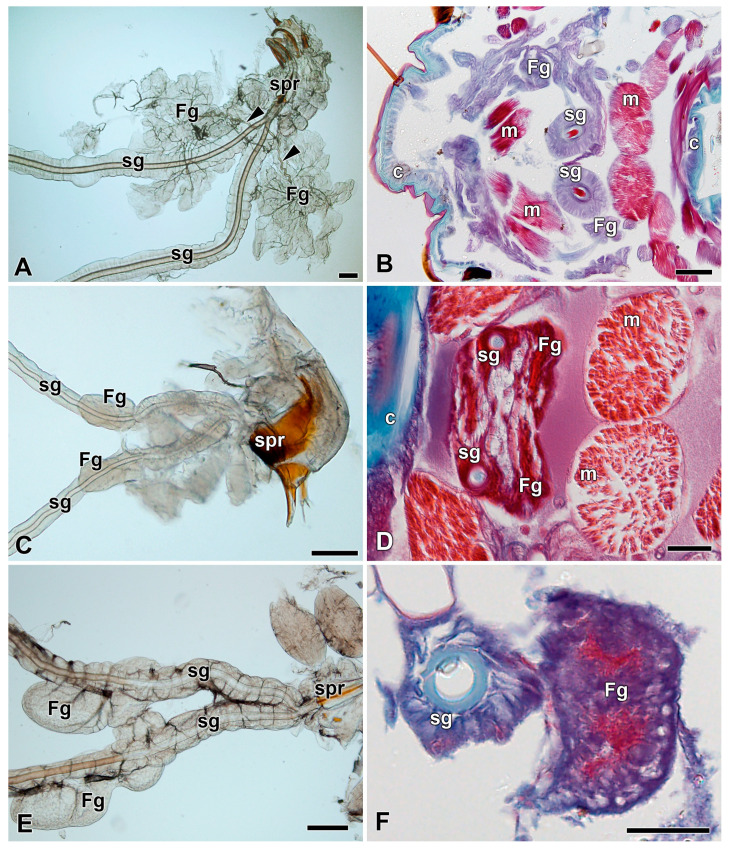
Spinning apparatuses in last instar larvae of the silkworm, *Bombyx mori*, and Pyralidae. (**A**,**B**) The silkworm, *Bombyx mori*. The arrowheads indicate the ducts of the FGs entering the lumen of the SGs. (**C**,**D**) The Mediterranean flour moth, *Ephestia kuehniella*. (**E**,**F**) The greater wax moth, *Galleria mellonella.* Brightfield images of whole-mount spinning apparatuses are shown in A, C, and E. The cross sections through the spinning apparatuses labeled with Masson trichrome stain in B, D, and F. Abbreviations: Cuticle (c), Filippi’s gland (Fg), muscles of the silk press (m), silk gland (sg), and silk press of the spinning apparatus (spr). Scale bars: A, C, and E, 100 µm; B, D, and F, 200 µm.

## Data Availability

Not applicable.

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
