# Peer review of "The Filippi’s Glands of Giant Silk Moths: To Be or Not to Be?"

_insects, 2021, doi:10.3390/insects12111040_

Round 1

Reviewer 1 Report

【Review comments】

The authors studied the presence (or not), morphology, anatomy and development of Filippi’s glands (FGs) for various silk moth and hawk moth species, and discussed whether the FGs is essential for spinning silk or cocoon construction or not. The experimental data shown here are all beautiful and therefore the one can catch well the discussed points from the data. Although the obtained knowledge is not enough to conclude the main question of authors, the present knowledge is believed to be important first materials. I recommend to accept this article with a few minor revision relating the following points.

(1)   In Introduction, it is not very clear the research aims and strategies, especially the reason why the species used in this study are selected. So, I suggest to mention the purposes and reasons for selection of species, described in each beginning sentences in sections of 3.1 (lines 128-131), 3.3 (lines 166-170), 3.4 (lines 185-188), 3.5 (lines 194-197). This would be helpful for readers.

(2)   I could not find explanation about “reproducibility” in each morphological observation. If it is not, please add the explanation. How many individual larvae did you examine in each species?

(3)   In this study, you concluded that the FGs are not essential for cocoon construction in the giant silk moths. Do you have any idea or strategy to clarify the function or role of FGs in the giant silk moths? If you have, I suggest to show it briefly (not in detail) as a future subject.

(4)   Present tittle seems not well represent the contents of manuscript.

(5)   I suggest to remove (or reduce) the words “surprisingly”, “interestingly”, and “markedly”.

Author Response

Reviewer #1:

The authors studied the presence (or not), morphology, anatomy and development of Filippi’s glands (FGs) for various silk moth and hawk moth species, and discussed whether the FGs is essential for spinning silk or cocoon construction or not. The experimental data shown here are all beautiful and therefore the one can catch well the discussed points from the data. Although the obtained knowledge is not enough to conclude the main question of authors, the present knowledge is believed to be important first materials. I recommend to accept this article with a few minor revision relating the following points.

Thank you!

  1. In Introduction, it is not very clear the research aims and strategies, especially the reason why the species used in this study are selected. So, I suggest to mention the purposes and reasons for selection of species, described in each beginning sentences in sections of 3.1 (lines 128-131), 3.3 (lines 166-170), 3.4 (lines 185-188), 3.5 (lines 194-197). This would be helpful for readers.

We added the aim of study and clarification of selection of the studied species to the Introduction part. (line 70-76):

„In the course of our previous study [22], we had noticed the absence of FGs in the larvae of the fifth instar of the cecropia silk moth, H. cecropia. To determine whether the absence of FGs is rather an exception in the superfamily Bombycoidae, we studied the morphology, anatomy, and development of FGs in selected silk moth and hawk moth species. We selected representatives of major saturniid and sphinghid clades based on current phylogenic tree [23]“.

We were also limited by the availability of the breeding material (www.wwb.co.uk).

  1. I could not find explanation about “reproducibility” in each morphological observation. If it is not, please adding the explanation. How many individual larvae did you examine in each species?

We thank reviewer for this comment, we added information about the reproducibility of morphological observations to the section of Material and methos (line 119):

„We used at least five experimental animals for each species examined.“ 

  1. In this study, you concluded that the FGs are not essential for cocoon construction in the giant silk moths. Do you have any idea or strategy to clarify the function or role of FGs in the giant silk moths? If you have, I suggest to show it briefly (not in detail) as a future subject.

We added following sentence at the end of the discussion. (line 255-256)

“Deciphering the exact function of FGs will require future additional molecular, proteomic, and metabolomic studies.” 

  1. Present tittle seems not well represent the contents of manuscript.

We believe that title agrees with the content of the manuscript.  

  1. I suggest removing (or reduce) the words “surprisingly”, “interestingly”, and “markedly”.

We have removed or reduced these words.

Reviewer 2 Report

Thank the editor for inviting me to review this paper, which reported the morphology of FGs in a range of wild (giant) silk moths and 19 several related species. After the comparison of presence and absence of the FGs in the silk-spinning and silk-none-spinning moth species, especial several species of giant silk moths and two hawk moth species that completely lack FGs 22 throughout their larval development. They claim that FGs are not critical for silk spinning, according to the evidence that some giant silk moths construct fully cocoons but lack of the FGs. These findings are interesting but only the morphological and anatomical evidences are barely enough to support this claim and conclusion. Thus, I can not recommend the present paper to be published in this journal.

  1. Zhang Yicheng et al. had reported the similar results in 1997 (Zhang Yicheng, Hao Dongtian, Liu Zengchao, Meng Ping, and Xu Fangguo, The Research on morphology of SG and FG of 18 species in lepidoptera, Journal of Shenyang Agricultural University, 1997-04, 28(2):104-108). In their study, 18 species among lepidoptera were chosen to perform morphological observation. They found morphological variance of FG between species, and they concluded that FG was a degenerating organ. Thus, the novelty of the present work is limited.
  2. Although some giant silk moths spin silk without FGs, is there any possibility that the morphology or some specific properties (maybe the mechanical properties, or the moisture absorption, or the adhesive properties) of the fiber spun by the moth without FGs have been lost? A more careful examination of the silk fibers produced by the species of FG-absent and FG-present is needed to understand the function of FG. Furthermore, could the morphological variation of FG correlate with the functional diversity of silk among these species?
  3. The authors mention in Line 243 that the species of the Bombycoidea superfamily possessed functional FGs. How to define the “functional FGs” or “non-functional FGs”, if the authors claim the FGs are not essential for silk spinning.
  4. Can the author explain what is the minor role of FGs in the maturation of silk fibers mentioned in the sentence in line 246-247 “the FGs are not crucial for cocoon construction and probably play some 246 minor role in maturation of silk fibers and/or resulting cocoon architecture.”
  5. There are many typing errors in the author information and supplementary figures sections.

Author Response

Reviewer #2:

Thank the editor for inviting me to review this paper, which reported the morphology of FGs in a range of wild (giant) silk moths and 19 several related species. After the comparison of presence and absence of the FGs in the silk-spinning and silk-none-spinning moth species, especial several species of giant silk moths and two hawk moth species that completely lack FGs 22 throughout their larval development. They claim that FGs are not critical for silk spinning, according to the evidence that some giant silk moths construct fully cocoons but lack of the FGs. These findings are interesting but only the morphological and anatomical evidences are barely enough to support this claim and conclusion. Thus, I cannot recommend the present paper to be published in this journal.

We believe that the lack of FGs in some silk moth species that still spin fully functional cocoon proves that that role of the FGs for the cocoon construction is not crucial. The exact role of the FGs, if any, will be a focus of our future studies on molecular, proteomic, and metabolomic levels. 

  1. Zhang Yicheng et al. had reported the similar results in 1997 (Zhang Yicheng, Hao Dongtian, Liu Zengchao, Meng Ping, and Xu Fangguo, The Research on morphology of SG and FG of 18 species in lepidoptera, Journal of Shenyang Agricultural University, 1997-04, 28(2):104-108). In their study, 18 species among lepidoptera were chosen to perform morphological observation. They found morphological variance of FG between species, and they concluded that FG was a degenerating organ. Thus, the novelty of the present work is limited.

We apologized but even through our extensive literature search about the FGs we did not find this article. Most likely because it does not mention Filippi´s glands in its title in full but only in abbreviation (FG). Moreover, this article is written in Chinese language, and it is impossible to download it from internet. Thus, we cannot make any conclusions based on this publication.

However, we would appreciate if the reviewer would be so kind and could provide us the PDF version of this article which might be useful for our future studies. 

  1. Although some giant silk moths spin silk without FGs, is there any possibility that the morphology or some specific properties (maybe the mechanical properties, or the moisture absorption, or the adhesive properties) of the fiber spun by the moth without FGs have been lost? A more careful examination of the silk fibers produced by the species of FG-absent and FG-present is needed to understand the function of FG. Furthermore, could the morphological variation of FG correlate with the functional diversity of silk among these species?

We thank for this comment. We believe that FGs might plays some minor roles in the properties of the silk fibers or the cocoon construction. Elucidating the precise function of FGs will be focus of our future studies.

  1. The authors mention in Line 243 that the species of the Bombycoideasuperfamily possessed functional FGs. How to define the “functional FGs” or “non-functional FGs”, if the authors claim the FGs are not essential for silk spinning.

We thank reviewer for this comment, we replaced word “functional with more appropriate “well developed” (line 251).  

  1. Can the author explain what the minor role of FGs in the maturation of silk fibers is mentioned in the sentence in line 246-247 “the FGs are not crucial for cocoon construction and probably play some 246 minor role in maturation of silk fibers and/or resulting cocoon architecture.”

We address this comment by adding following sentence to the end of the Discussion section.

“To decipher the exact function of FGs will require future additional molecular, proteomic, and metabolomic studies.”

  1. There are many typing errors in the author information and supplementary figures sections.

We have corrected the typing errors in the mentioned manuscript section.

Reviewer 3 Report

Dear Authors,

This is an intersting and overall well written paper, especially interesting for those who are much interested in silk production physilogy of certain moth groups. I have some minor comment only, which can be identified in the sticky notes boxes.

I find it confusing that the species in M and M are in alphabetical order, while in Result and Discussion are grouped taxonomically. I would suggest to follow the same logic to easier follow the content. Moreover the Supplementary material should be clearly separated and references should be supplied with an "S" on figure legends. In Discussion I suggest more references back to figures.

A major point I miss that in the ms the authors mention that they provide a comparison between selected moths of saturniids and sphingids which allows correlation between FG morphology and their phylogenic position, however the speculation on their function is very limited in the work.

So despite the high quality of the morphological work I think it is an avarage publication without significant breakthrough, but worth publishing.

Best regards, A Reviewer

Author Response

Reviewer #3:

Dear Authors,

This is an interesting and overall well written paper, especially interesting for those who are much interested in silk production physiology of certain moth groups. I have some minor comment only, which can be identified in the sticky notes boxes.

I find it confusing that the species in M and M are in alphabetical order, while in Result and Discussion are grouped taxonomically. I would suggest to follow the same logic to easier follow the content. Moreover, the Supplementary material should be clearly separated and references should be supplied with an "S" on figure legends. In Discussion I suggest more references back to figures.

We thank for this comment. We are aware of different ordering of species in the Material and methods and the Results and Discussion. We use the alphabetical order in the Material and methods section for the reader to easily find out the specification of each used species in this study. In Results and Discussion, the taxonomical grouping of species is logical for the flow of the paper.

We added more references to Figures in the Discussion section as suggested by the reviewer.    

A major point I miss that in the ms the authors mention that they provide a comparison between selected moths of saturniids and sphingids which allows correlation between FG morphology and their phylogenic position, however the speculation on their function is very limited in the work.

Based on the study we cannot extensively speculate about the precise function of the FGs. To decipher the exact function of FGs will require future additional molecular, proteomic, and         metabolomic studies.

So despite the high quality of the morphological work I think it is an average publication without significant breakthrough, but worth publishing.

Thank you!

Reviewer 4 Report

The paper is well written and the comparison between the Filippi’s glands in different Lepidoptera is an interesting topic since it looks like a non-essential accessory organ for another type of glands which has an incredibly high level of specialization like the silk gland. The paper is basically ready to be published. I just noted a few minor issues/typos.

Line 9             “possess silk glands” without "the"

Line 20            “Our results confirm that the majority of studied silk moths...”

Line 21             “Surprisingly”

Line 71              “are absent in a number of”

Line 118         maybe a reference for Paraplast?

In the text there are references to figures included as Supplementary materials, I guess, but only Figure S5 is referenced correctly (I mean with an S) while the other ones look like double. Please insert the S in captions where needed.

In some supplementary figures there is no definition for arrowheads and in Figure 4S the caption says “Arrows depict the FGs” but I can’t see any arrow. Please double check this.

Lines 170 and 219      I would rather use “similarly” instead of “similar”

Author Response

Reviewer #4:

The paper is well written and the comparison between the Filippi’s glands in different Lepidoptera is an interesting topic since it looks like a non-essential accessory organ for another type of glands which has an incredibly high level of specialization like the silk gland. The paper is basically ready to be published. I just noted a few minor issues/typos.

Line 9             “possess silk glands” without "the"

Line 20            “Our results confirm that the majority of studied silk moths...”

Line 21             “Surprisingly”

Line 71              “are absent in a number of”

Line 118         maybe a reference for Paraplast?

In the text there are references to figures included as Supplementary materials, I guess, but only Figure S5 is referenced correctly (I mean with an S) while the other ones look like double. Please insert the S in captions where needed.

In some supplementary figures there is no definition for arrowheads and in Figure 4S the caption says “Arrows depict the FGs” but I can’t see any arrow. Please double check this.

 Lines 170 and 219      I would rather use “similarly” instead of “similar”

We thank the reviewer for kind comments on our manuscript. We appreciate reviewer for careful reading. We have corrected all issues/typos noted. We provided the reference for Paraplast. We added “S” to the supplementary figure’s captions, we corrected the problems with definition of arrows and arrowheads in the supplementary figures as suggested.

Round 2

Reviewer 2 Report

The quality of the revised manuscript is well improved, and the authors also well dressed my major concerns, thus I think the current version is suitable for publication in Insects.

Author Response

Dear Reviewer,

Please find enclosed revised manuscript: “The Filippi’s Glands of Giant Silk Moths: To Be or Not to Be?”

We have added requested citation of two articles to the appropriate places in the manuscript and in the list of the References.

We are grateful for your valuable comments 
